# Quantitative Analysis of Natural and Anthropogenic Factors Influencing Vegetation NDVI Changes in Temperate Drylands from a Spatial Stratified Heterogeneity Perspective: A Case Study of Inner Mongolia Grasslands, China

**Shengkun Li [1,2], Xiaobing Li [1,2,\*], Jirui Gong [1,2], Dongliang Dang [2], Huashun Dou [2] and Xin Lyu [2]**

[1] State Key Laboratory of Earth Surface Process and Resource Ecology, Beijing Normal University, Beijing 100875, China; sklee@mail.bnu.edu.cn (S.L.); jrgong@bnu.edu.cn (J.G.)

[2] Faculty of Geographical Science, Beijing Normal University, Beijing 100875, China; 201621190009@mail.bnu.edu.cn (D.D.); hsdou@mail.bnu.edu.cn (H.D.); lyuxin@bnu.edu.cn (X.L.)

\* Correspondence: xbli@bnu.edu.cn; Tel.: +86-010-58807212

**Abstract:** The detection and attribution of vegetation dynamics in drylands is an important step for the development of effective adaptation and mitigation strategies to combat the challenges posed by human activities and climate change. However, due to the spatial heterogeneity and interactive influences of various factors, quantifying the contributions of driving forces on vegetation change remains challenging. In this study, using the normalized difference vegetation index (NDVI) as a proxy of vegetation growth status and coverage, we analyzed the temporal and spatial characteristics of the NDVI in China's Inner Mongolian grasslands using Theil–Sen slope statistics and Mann–Kendall trend test methods. In addition, using the GeoDetector method, a spatially-based statistical technique, we assessed the individual and interactive influences of natural factors and human activities on vegetation-NDVI change. The results show that the growing season average NDVI exhibited a fluctuating upward trend of 0.003 per year from 2000 to 2018. The areas with significant increases in NDVI ($p < 0.05$) accounted for 45.63% of the entire region, and they were mainly distributed in the eastern part of the Mu Us sandy land and the eastern areas of the Greater Khingan Range. The regions with a decline in the NDVI were mainly distributed in the central and western regions of the study area. The GeoDetector results revealed that both natural and human factors had significant impacts on changes in the NDVI ($p < 0.001$). Precipitation, livestock density, wind speed, and population density were the dominant factors affecting NDVI changes in the Inner Mongolian grasslands, explaining more than 15% of the variability, while the contributions of the two topography factors (terrain slope and slope aspect) were relatively low (less than 2%). Furthermore, NDVI changes responded to the changes in the level of specific influencing factors in a nonlinear way, and the interaction of two factors enhanced the effect of each singular factor. The interaction between precipitation and temperature was the highest among all factors, accounting for 39.3% of NDVI variations. Findings from our study may aid policymakers in better understanding the relative importance of various factors and the impacts of the interactions between factors on vegetation change, which has important implications for preventing and mitigating land degradation and achieving sustainable pasture use in dryland ecosystems.

**Keywords:** NDVI; vegetation dynamics; influencing factors; spatial stratified heterogeneity; geographical detector method

## 1. Introduction

Drylands, covering about 41.30% of the Earth's terrestrial surface and supporting more than 38% of the global population [1], are characterized by a lack of water, infertile soil, and high climate variability. They are highly susceptible to climate fluctuations and

human activities [2,3]. Because of the limitations imposed by water resource availability and challenging climate change effects, drylands fall victim to persistent land degradation problems that have led to the desertification of 3.6 billion hectares worldwide and have threatened the lives and livelihoods of the local people [3,4]. Monitoring land degradation and identifying its potential causes are of great significance to sustainable land use. As the primary producer in the ecosystem, the ground vegetation links the carbon–water cycle and the energy flow within the hydrosphere, pedosphere, and atmosphere [5,6], and it plays a fundamental role in providing ecosystem goods and maintaining terrestrial ecosystem functions [7]. The vegetation conditions of degraded land have always been used as a proxy to quantitatively detect ecosystem processes at both local and regional scales [8–11]. With the help of satellite remote sensing images, the detection and attribution of vegetation greening and browning trends have emerged as a popular subject in the scientific community over the past several decades [12], and the relation between the normalized difference vegetation index (NDVI) and vegetation growth status and coverage has been well established. Due to the spatial heterogeneity and the combined effects of the driving factors [13,14], quantifying the contributions of the main drivers of vegetation change remains challenging. It is urgent that techniques be developed to help disentangle the contributions of factors to variations in vegetation for the development of strategies for vegetation restoration and desertification prevention in drylands.

In general, vegetation change was influenced by intertwined natural and human-induced factors. The impact of global climate change on vegetation growth is a major research priority. Numerous studies have been carried out related to the response of the NDVI to variations in climatic factors (e.g., air temperature, solar radiation, and precipitation) at different spatiotemporal scales [5,6,15], aimed at improving our knowledge of the mechanistic link between the effects of climate change on vegetation activity. Over the last decades, human activities became diverse and intensive, exerting greater pressure on terrestrial ecosystems [3,16]. Anthropogenic factors manifest primarily in land-use change (LUCC) or changes in management measures [17,18]. Urbanization, characterized by the occupation of vegetation-covered surfaces by impervious ground, may lead to vegetation degradation [19]. Overgrazing, cultivation of arable land, and deforestation have resulted in bare ground and soil erosion, which may result in vegetation degradation [20], while the enclosure management of degraded rangeland may promote vegetation restoration [21].

The residual trend method (RESTREND), ecosystem modeling methods, and various mathematical models are widely used for quantifying the influences of driving factors on changes in vegetation growth. The RESTREND method distinguishes between human-induced and climate-driven vegetation changes based on the trend of NDVI residues (defined as the differences between the actual and predicted NDVI values) [22], and it is predominantly useful in studies of regions where water is limiting [9,23,24]. However, the RESTREND method is associated with some uncertainties [25]. The results of the RESTREND method may vary considerably with the time employed to compute the NDVI-precipitation regression and the trends of its residuals [23,26]. Moreover, this method attributes the residuals to the total effects of all human disturbances, making it difficult to disentangle and compare the contributions of different human activities on vegetation variations [18]. The mathematical models mainly include regression, correlation analysis, and the structural equation modeling method [6,18,19,27]. Most of the mathematical models detect the impacts of the environmental variables on the vegetation dynamics using a linear hypothesis [28]. However, theory and empirical evidence suggest that the trajectory of the responses of the vegetation index to the influencing factors is often nonlinear [2,29,30], so the linearity assumptions may result in erroneous conclusions and misleading interpretations. Many process-based ecosystem models have been developed to quantify the responses of environmental variables to key ecological processes in a nonlinear way [28,31,32], overcoming the deficiencies of the mathematical models. However, ecosystem modes usually require a large number of inputs and parameter settings, and there are uncertainties in the models' structures and parameter choices [33], which may lead to

inconsistent model results [15]. The GeoDetector method, which was developed by Wang in 2010 [34], quantifies the impacts of factors on geographical phenomena or attributes from a spatially stratified heterogeneity perspective [34,35]. The GeoDetector method does not involve complex parameter settings, nor does it follow the restrictive assumptions of traditional statistical methods. This technique has been used to evaluate the influences of factors in the eco-environmental and social science fields [15,36–40]. The GeoDetector method can be a promising tool for exploring the associations between various impact factors and vegetation changes in drylands.

With climate change and increasing anthropogenic activities, the vulnerable ecosystems of the drylands in northern China have been degraded to varying degrees, posing severe ecological and environmental problems [41–43]. In order to reverse the environmental degradation trend, particularly in the ecologically fragile regions, several ecological conservation programs were carried out in the late 1990s [44,45]. Since then, land-use patterns have changed substantially [45]. An in-depth understanding of the spatial features and the changes in and underlying the driving mechanisms of vegetation activity is important to improve policymakers' understanding of the sustainable use of vegetation resources and for the development of reasonable strategies for ongoing ecological restoration. At present, scholars have mainly focused on the relationships between vegetation variations and climatic factors at different time and spatial scales in the drylands of northern China [15,24,26,46–49], but they have paid little attention to different human activities (e.g., grazing pressure, land use conversions). In addition, few studies have considered the potential of interactive effects between the factors impacting vegetation changes. If the interactions between factors are not taken into consideration, the results may be biased [19].

In this study, the Inner Mongolian grasslands were selected as the study area. The Inner Mongolian grasslands act as an ecological protective belt for eastern China and provide plenty of ecosystem services (e.g., food supply, grass production, climate regulation, carbon sequestration, soil erosion control, and cultural heritage) [50]. The objectives of this work were twofold: (1) to investigate the temporal and spatial characteristics of the NDVI in the Inner Mongolian grasslands from 2000–2018; and (2) to examine the individual contributions and interactive effects of natural factors and human activities on vegetation changes using the GeoDetector method. This work aims to provide a scientific foundation for detecting the underlying mechanism of vegetation changes in temperate grasslands.

## 2. Materials and Methods

### 2.1. Study Area

The Inner Mongolian grasslands, located in the drylands of northern China (105°18′–125°15′E, 37°38′–50°50′N; Figure 1a), cover an area of approximately $78.2 \times 10^4$ km$^2$. This region is mainly arid and semi-arid [48]. It is ecologically fragile and is vulnerable to climate variations and human activities, but it plays a critical role in safeguarding the ecological security of northern China's agricultural plain and metropolitan regions [44]. Dominated by a temperate continental climate, the Inner Mongolian grasslands have annual precipitation of 120–520 mm and an annual mean temperature of −2 to 10 °C. More than 80% of the annual total precipitation is concentrated between July and September, which coincides with the period of high temperatures [44]. The climate has distinct seasonal characteristics: (1) windy and dry in spring, with strong evaporation; (2) warm and hot in summer, with an uneven precipitation distribution; (3) a short autumn, with early frost and snow and large day–night temperature differences; and (4) a cold and long winter. The vegetation across the Inner Mongolian grasslands has obvious east–west zonal distribution characteristics. From west to east, the precipitation and soil fertility gradually increase, and the solar radiation gradually decreases, forming three different types of temperate grasslands (Figure 1d). There are also nonzonal vegetation types, including saline meadows and marshes along riverbanks and large tracts of sandy lands, which are closely associated with site-specific geographical characteristics (e.g., water bodies, topography, and salinization). The study area is dominated by high plains and low mountains, which are characterized

by high elevations in the central and western areas and low elevations in the southeastern and northern areas, with altitudes ranging from 90 m to 2300 m (Figure 1b).

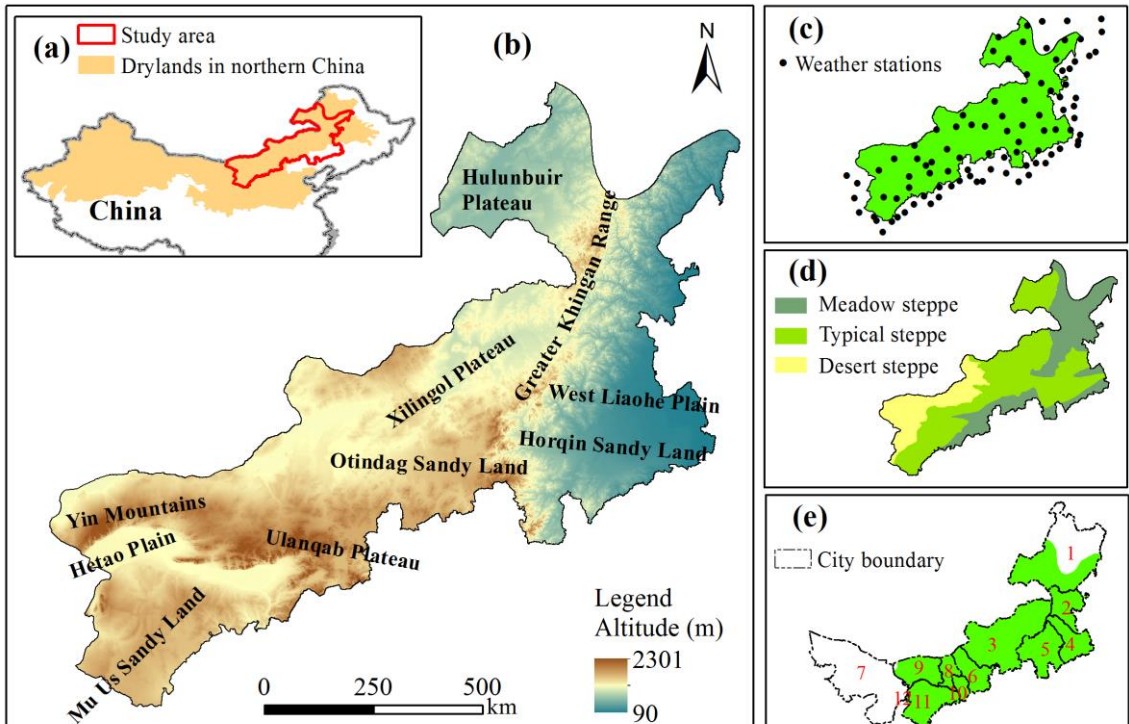

**Figure 1.** (**a**) Location of the study area, (**b**) topographical conditions, (**c**) weather station distribution, (**d**) ecoregions of the study area, and (**e**) city boundaries. The numbers in (**e**) represent the prefecture-level cities (1—Hulunbuir; 2—Hinggan; 3—Xilingol; 4—Tongliao; 5—Chifeng; 6—Ulanqab; 7—Alxa; 8—Baotou; 9—Bayannur; 10—Hohhot; 11—Erdos; 12—Wuhai).

## 2.2. Data Acquisition and Processing

The NDVI values were extracted from the MOD13A2 product (Version 6, 1000 m resolution, 16-days composite). The MOD13A2 product was geographically projected using the MODIS Reprojection Tool (MRT) software; then, the maximum value composite method (MVC) was used to obtain monthly NDVI data to reduce the effects of clouds and image noise. The MVC method still cannot guarantee that all pixels of an image are cloud-free. In this study, we used the variable weight filtering method proposed by Zhu to reconstruct a set of high-quality NDVI time-series data; the reconstructed vegetation index time-series data can enhance the application capability of vegetation index time-series data in the study of vegetation–climate factor interactions [51]. Given that most of the plants withered and stopped growing during the winter, we used the growing season (defined as April to October) NDVI to detect the inter-annual variations in the vegetation activity [48,49].

The meteorological datasets covering the period of 2000–2018 were compiled from ninety-six weather stations (Figure 1c). The datasets included daily values of the mean temperature, precipitation, sunshine duration, relative air humidity, and mean wind speed, which were obtained from the National Meteorological Information Center of China. The Solar Energy Resource Evaluation method (QX/T 89-2008), which was developed by the China Meteorological Administration, was used to estimate solar radiation. The spatial distribution results for meteorological station data at a spatial resolution of 1000 m were obtained via spatial interpolation using ArcGIS 10.2.

The 2000 and 2015 land cover type data (1000 m resolution) were retrieved from the Resource and Environment Science and Data Center. The dataset was interpreted visually based on Landsat thematic mapper images and unmanned aerial vehicles, which are char-

acterized as being highly accurate via random sampling checks and field surveys. The data contain 26 secondary categories with a comprehensive evaluation accuracy of >90% [52].

The soil type data were extracted from the soil map of China (1:1,000,000), which was provided by the Chinese Soil Census Office. The data were compiled by soil generation classification standards.

The topographic data consisted of altitude, slope, and aspect data. Through image mosaicking, we obtained a DEM of the study area with a spatial resolution of 90 m from the Geospatial Data Cloud site (http://www.gscloud.cn, accessed on 25 March 2022).

The administrative boundaries, roads, and settlements (1:250,000) vector data were obtained from the National Catalogue Service for Geographic Information, Ministry of Natural Resources of China.

The statistical data, including the total population, gross regional product, agricultural mechanical power, fertilizer applied for agriculture, grain production, oil production, and quantity of livestock (including goats, sheep, horses, cattle, and camels), at the county level, were obtained from the Inner Mongolia Statistical Yearbook (https://data.cnki.net/, accessed on 20 March 2022). According to prior research, we used an equivalent unit of grazing (i.e., "sheep unit") to normalize the grazing intensity among different species [53]. Using the empirical formula [10,53], we set the transition factor for large livestock (e.g., cattle, donkeys, camels, and horses) to 6, whereas the transition factor was set to 1 for goats and sheep.

### 2.3. Mann-Kendall Trend Test and Sen's Slope Estimator

In this work, the Sen trend analysis and the Mann–Kendall test [54–56] were used to detect the trend slopes and significance of trends in the NDVI time series, respectively. The procedure for the nonparametric Mann–Kendall trend test [55,56] is as follows:

$$S = \sum_{i=1}^{n-1} \sum_{j=i+1}^{n} \text{sgn}(x_j - x_i) \tag{1}$$

In Equation (1), S denotes the standardized test statistic value, $x_i$ and $x_j$ are data values at time i and j, respectively; n is the length of time series; and $\text{sgn}(x_j - x_i)$ is the sign function, which is calculated as follows:

$$\text{sgn}(x_j - x_i) = \begin{cases} -1, & if\ x_j - x_i < 0 \\ 0, & if\ x_j - x_i = 0 \\ +1, & if\ x_j - x_i > 0 \end{cases} \tag{2}$$

In this study, the length of time series n = 19, and the trend test were conducted using the $Z_S$ value, which is defined as follows:

$$Z_S = \begin{cases} \dfrac{S+1}{\sqrt{\text{Var(S)}}}, & if\ S < 0 \\ 0, & if\ S = 0 \\ \dfrac{S-1}{\sqrt{\text{Var(S)}}}, & if\ S > 0 \end{cases} \tag{3}$$

In Equation (3), the variance Var(S) is computed as:

$$\text{Var(S)} = \frac{n(n-1)(2n+5) - \sum_{i=1}^{m} t_i(t_i-1)(2t_i+5)}{18} \tag{4}$$

In Equation (4), m is the number of tied groups in the time series and $t_i$ is the width of the tied groups. In this study, a significance level of $\alpha = 0.05$ was used. It is assumed that, for null hypothesis, the data are arranged with no significant trend. When $|Z| > Z_{1-\alpha/2}$, the null hypothesis is rejected and the trend of the change in the series data is considered to be significant.

The Sen slope calculation is carried out as:

$$\beta = \text{Median}\left(\frac{x_j - x_i}{j - i}\right) \tag{5}$$

where Median() denotes the median function of the requested series; $\beta$ is the slope of the time series x; and a negative $\beta$ value indicates a decreasing trend in the series.

### 2.4. GeoDetector Method

Spatial stratified heterogeneity (SSH), referring to a within sub-region variance of less than that between the sub-regions [35], is ubiquitous in ecological phenomena, such as soil types, land use types, and climate zones. The GeoDetector comprises a series of spatial statistical methods, and it is frequently used to detect the SSH of the dependent variables without linear assumptions and reveal the driving forces behind a phenomenon by quantifying the impact of associated factors. The GeoDetector assumes that if an independent variable (e.g., precipitation) has a certain degree of influence on a dependent variable (i.e., NDVI changes), then the spatial patterns of the two variables have high similarity. Figure 2 illustrates the principle of the GeoDetector; spatial variances within each sub-region and among the different sub-regions are compared to identify the explanatory powers of the potential explanatory variables [34,39,57].

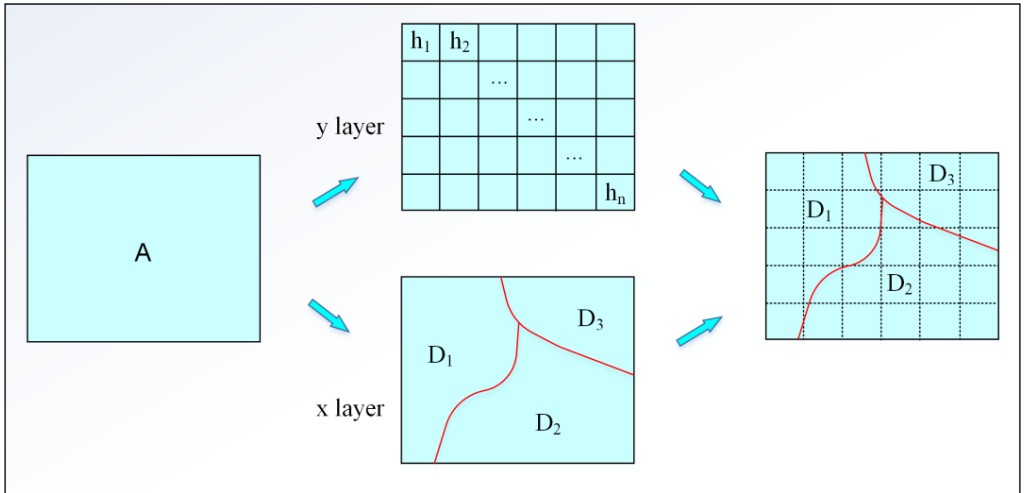

**Figure 2.** Illustration of the principle of the GeoDetector model. The study area A, the grid system H = {h$_i$, i = 1, 2, . . . , n}, and the sub-region of the potential factor D = {D$_i$, i = 1, 2, 3}.

#### 2.4.1. Single Factor Influence Detection

The impact of an individual factor on changes in NDVI can be measured using the $q$-statistic [34,35]:

$$q = 1 - \frac{\sum_{h=1}^{L} N_h \sigma_h^2}{N\sigma^2} \tag{6}$$

where $q$ is the measurement index of the factor. The range of $q$-statistic is [0, 1]. Based on the model principle, the larger the $q$-statistic is, the stronger the independent variable represents the heterogeneity of the dependent variable. $L$ refers to the number of stratifications of factor X; $N_h$ and $N$ are the numbers of units in sub-region h and over the whole study region, respectively; and $\sigma^2$ and $\sigma_h^2$ represent the variances of variable Y over the entire study region and in sub-region $h$, respectively.

#### 2.4.2. Interaction Detection of Pairwise Factors

The interactive impact of two explanatory factors (X1 and X2) on NDVI change also can be quantified by $q$-statistic. The module of interaction detection quantifies the interaction between two factors by comparing $q$(X1∩X2) with $q$(X1) and $q$(X2) to assess whether the

factors weaken or enhance one another or are independent of each other, in which $q(X1 \cap X2)$ indicates the explanatory power of a new factor created by overlaying the layer of the two variables in GIS tools (Supplementary Figure S1). Generally, the results of the interaction detector encompass five categories (Figure 3).

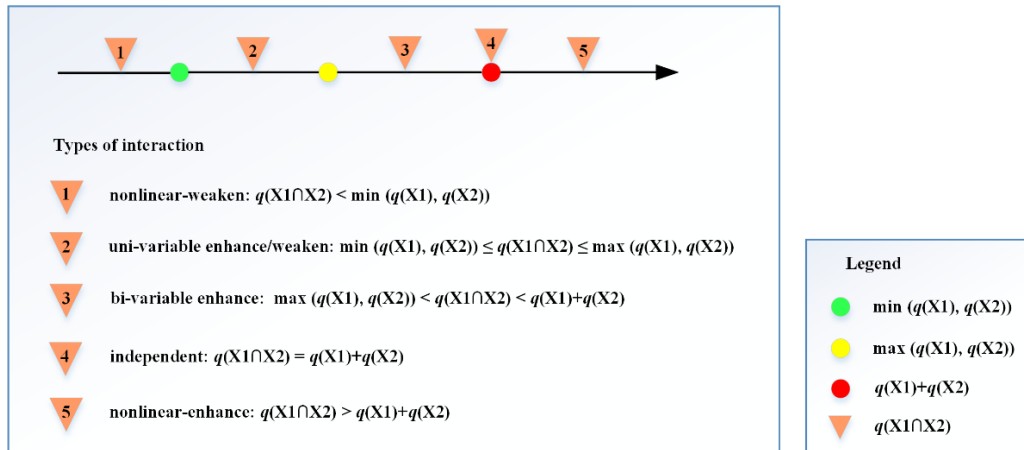

**Figure 3.** Judgment for interaction types between explanatory variables. Note: max() and min() denote the maximum and minimum functions, respectively. $q(X1 \cap X2)$ represents the interaction between factors X1 and X2. Modified from prior research [38,57,58].

2.4.3. Selection of Factors

In this work, we chose the slope of the change in the NDVI from 2000–2018 as the dependent variable and selected 15 potential natural and human factors (Table 1, Figure 4). Specifically, in addition to the climatic factors, we included soil and topography as fundamental environmental factors, which have been demonstrated to be critical to inter-annual variations of vegetation [9,59–61]. Additionally, six factors (road impact, geographical location, population pressure, grazing pressure, land use/cover change, and economic development) were selected to reflect the magnitude of anthropogenic influences [19,59]. In this study, we reclassified land cover types into six types, and the land cover maps (water bodies were excluded) for 2000 and 2015 were superimposed to generate a land use/cover change (LUCC) map. The spatial distribution of the grades for all driving factors can be found in Figure 4.

**Table 1.** Potential driving factors of vegetation variation in the study area.

| Category | Index | Abbreviation | Unit |
|---|---|---|---|
| Climate | Annual precipitation | Pre | mm |
| | Annual mean temperature | Tem | °C |
| | Annual solar radiation | SR | MJ·m$^{-2}$ |
| | Annual mean wind speed | WS | m·s$^{-1}$ |
| | Annual mean relative air humidity | RH | % |
| Topography | Altitude | Alt | m |
| | Terrain slope | Slopd | ° |
| | Slope aspect | Slopa | ° |
| Soil | Soil type | Soilt | categorical |
| Human activity | Distance to the nearest road | DNR | km |
| | Distance to the nearest county centers | DNC | km |
| | Population density | Popd | person·km$^{-2}$ |
| | Per capita gross regional product | GRP | 10,000 yuan person$^{-1}$ |
| | Livestock density | Livstd | sheep·km$^{-2}$ |
| | Land use/cover change | LUCC | categorical |

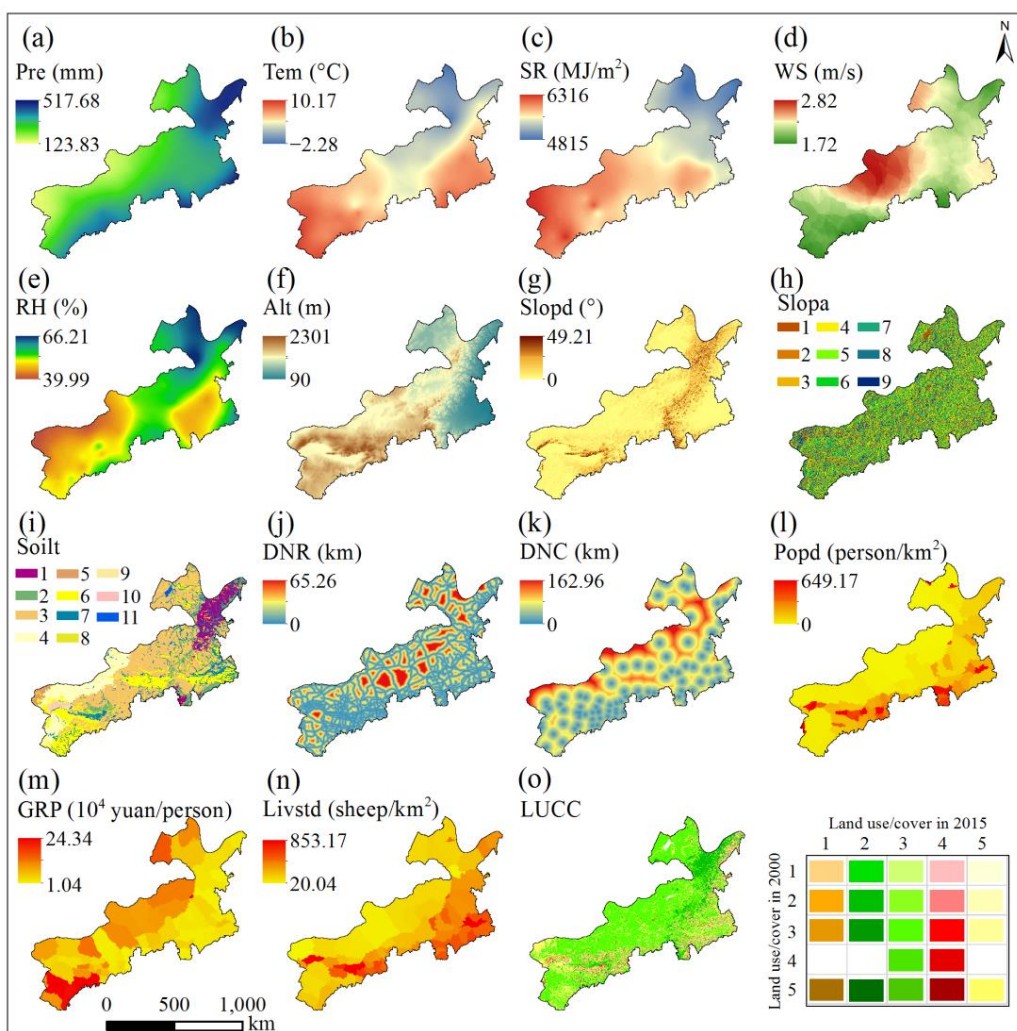

**Figure 4.** The spatial distributions of all factors. The numbers in the legend of (**h**) represent (1) Flat ground, (2) North slope, (3) Northeast slope, (4) East slope, (5) Southeast slope, (6) South slope, (7) Southwest slope, (8) West slope, and (9) Northwest slope. The numbers in the legend of (**i**) represent (1) Luvisols, (2) Semi-luvisols, (3) Caliche soils, (4) Arid soils, (5) Desert soils, (6) Skeletol primitive soils, (7) Semi-hydromorphic soils, (8) Hydromorphic soils, (9) Saline soils, (10) Anthrosols, and (11) Others. The numbers in the legend of (**o**) represent (1) Cropland, (2) Forest, (3) Grassland, (4) Construction land, and (5) Unused land. Pre: precipitation; Tem: air temperature; SR: solar radiation; WS: wind speed; RH: relative air humidity; Alt: altitude; Slopd: terrain slope; Slopa: slope aspect; Soilt: soil type; DNR: distance to the nearest road; DNC: distance to the nearest county centers; Popd: population density; GRP: Per capita gross regional product; Livstd: livestock density; LUCC: land use/cover change.

### 2.4.4. Factor Grading Optimization in the GeoDetector Method

Since the GeoDetector method is only suitable for dealing with discrete or categorical variables, all the continuous predictor variables should be discretized using appropriate discretization methods before modeling [38,62]. In this study, the twelve factors, namely, five post-interpolation meteorological factors, two topography factors (altitude and terrain slope), and five anthropogenic factors (distance to the nearest road, distance to the nearest county centers, population density, per capita gross regional product, and livestock density), are continuous variables. We converted the twelve continuous variables into discrete ones.

To reduce the subjectivity of user-defined discretization and ensure the best-quality modelling results, the optimal discretization methods were determined from five types of unsupervised discretization methods, including geometrical interval (GI), natural breaks

(NB), equal interval (EI), quantile (QU), and standard deviation (SD) methods [38,62,63]. The procedures used for the factor grading optimization are as follows (Figure 5). First, we classified each continuous variable based on the five discretization methods and fourteen levels (stratification numbers of 2 to 15). Then, we extracted the values of the NDVI tendency layer and all of the classification layers. Finally, we calculated the $q$-statistics of each continuous predictor variable in all of the classification cases and plotted them to show their changes (Figure 6).

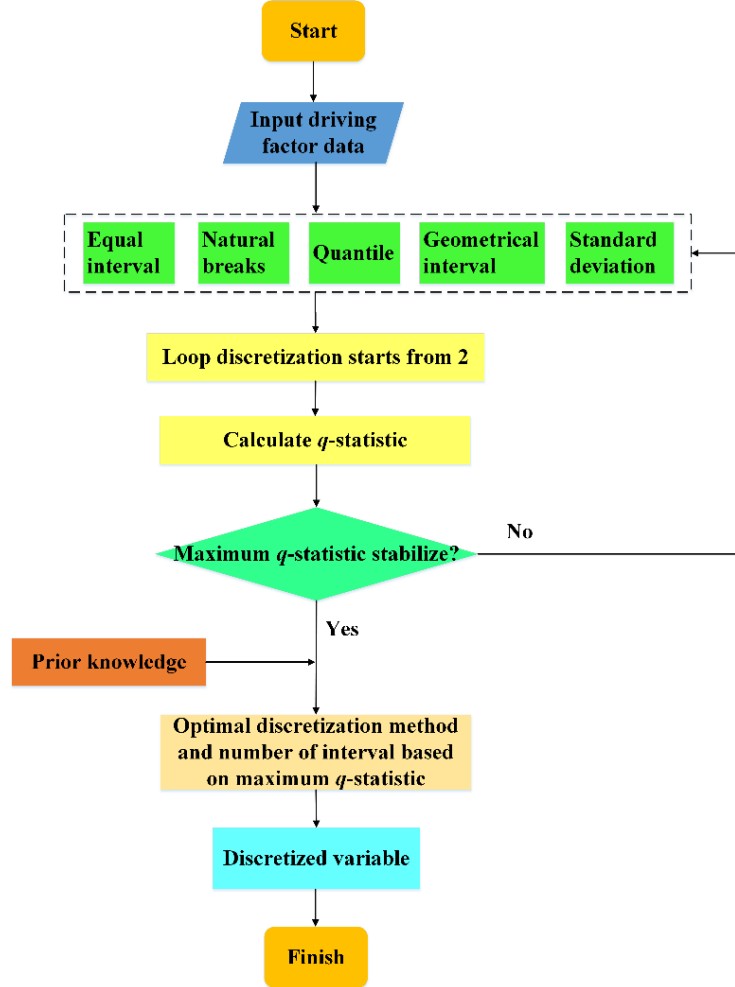

**Figure 5.** Flowchart illustrating the process of determining the optimal discretization method and stratification number.

A combination of classification algorithms with prior knowledge was needed to classify the continuous variable when using the GeoDetector method [34,57]. As shown in Figure 6, the maximum $q$-statistics of the factors generally increased as the number of stratifications increased. When the number of stratifications reached a certain value, the maximum $q$-statistic stabilized. This certain value was defined as the stable value. When the stratification number was bigger than the stable value, the characterization identified by GeoDetector remained unchanged, implying that more discretization intervals do not mine the information of the continuous variables. Considering the maximum stratification numbers (7, 8, 7, and 9, respectively) used in relevant studies [38,64–66] and the stable value observed in this study (approximately 10 in Figure 6), we limited the maximum number of stratifications to 10. The largest $q$-statistic value indicates the optimal discretization method and stratification number [38,62,63]. Based on this principle, we determined the optimal discretization methods and stratification numbers of each predictor variable. The impact

factors with the optimal discretization methods and stratification numbers can be found in Table 2.

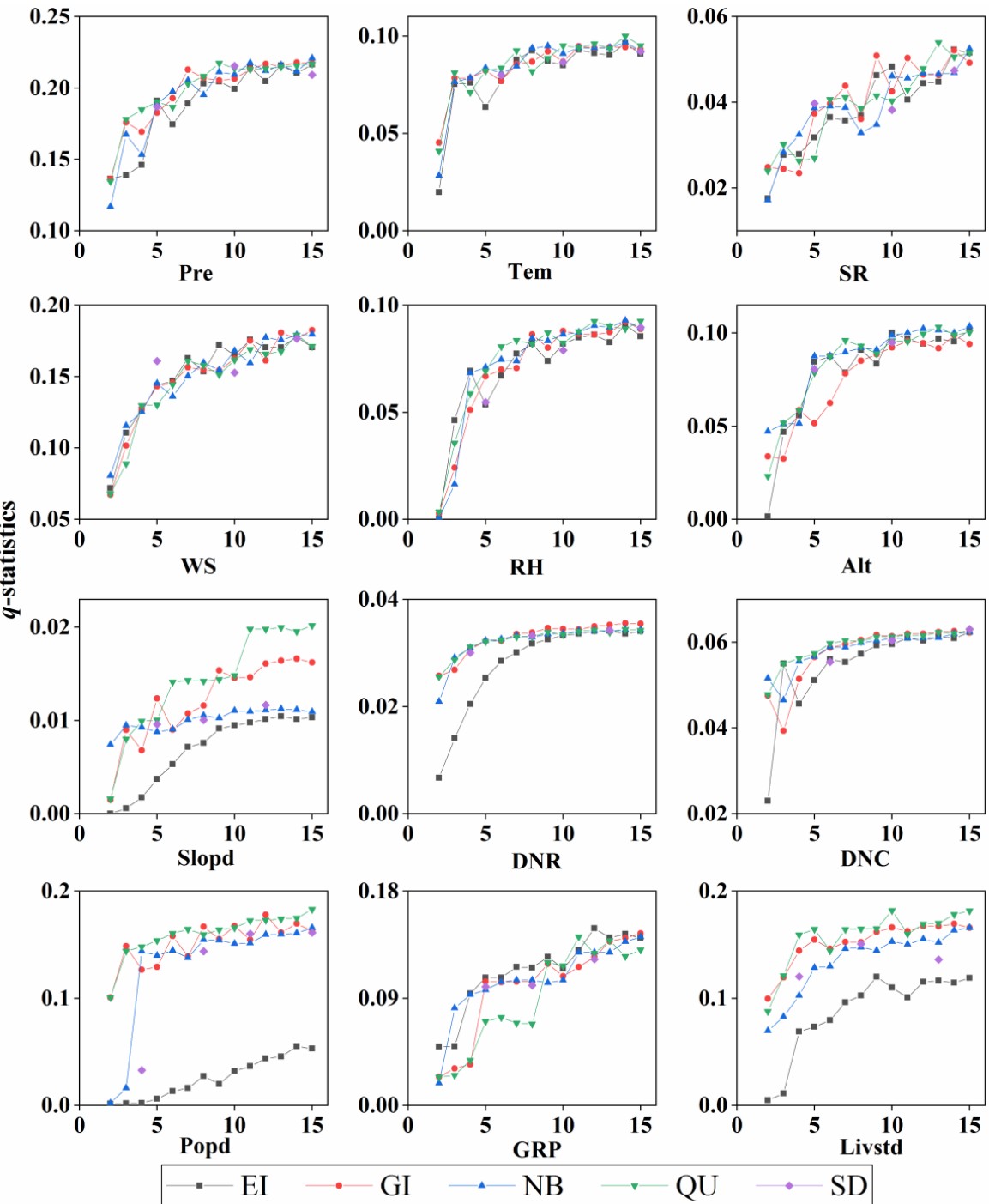

**Figure 6.** Comparison of the *q*-statistics under the different discretization methods and stratification number combinations. The five types of discretization methods include equal interval (EI), geometrical interval (GI), natural break (NB), quantile (QU), and standard deviation (SD). Pre: precipitation; Tem: air temperature; SR: solar radiation; WS: wind speed; RH: relative air humidity; Alt: altitude; Slopd: slope; DNR: distance to the nearest road; DNC: distance to the nearest county centers; Popd: population density; GRP: Per capita gross regional product; and Livstd: livestock density.

**Table 2.** The classification of potential driving factors. The units of Pre, Tem, SR, WS, RH, Alt, Slopd, DNR, DNC, Popd, GRP, and Livstd are mm, °C, MJ·m$^{-2}$, m·s$^{-1}$, %, m, °, km, km, person·km$^{-2}$, 10,000 yuan person$^{-1}$, and sheep·km$^{-2}$, respectively. In the parentheses of the table header, QU, EI, and GI correspond to three discretization methods of the quantile, equal interval, and geometrical interval, respectively, and the numbers represent the number of stratifications.

| Category/Factor | Pre (QU-9) | Tem (QU-10) | SR (GI-9) | WS (EI-9) | RH (GI-10) | Alt (EI-10) | Slopd (GI-9) |
|---|---|---|---|---|---|---|---|
| 1 | 123.8–194.9 | −2.28 to 0.74 | 4815–5130 | 1.72–1.85 | 40.0–43.7 | 90–311 | 0–0.17 |
| 2 | 194.9–233.5 | 0.74–2.11 | 5130–5371 | 1.85–1.97 | 43.7–46.5 | 311–532 | 0.17–0.24 |
| 3 | 233.5–269.0 | 2.11–3.18 | 5371–5555 | 1.97–2.09 | 46.5–48.5 | 532–753 | 0.24–0.40 |
| 4 | 269.0–306.1 | 3.18–3.97 | 5555–5695 | 2.09–2.21 | 48.5–50.0 | 753–974 | 0.40–0.79 |
| 5 | 306.1–332.3 | 3.97–4.94 | 5695–5802 | 2.21–2.33 | 50.0–51.1 | 974–1195 | 0.79–1.71 |
| 6 | 332.3–352.4 | 4.94–5.72 | 5802–5884 | 2.33–2.45 | 51.1–52.7 | 1195–1416 | 1.71–3.88 |
| 7 | 352.4–378.7 | 5.72–6.89 | 5884–5991 | 2.45–2.58 | 52.7–54.7 | 1416–1638 | 3.88–8.97 |
| 8 | 378.7–418.8 | 6.89–7.48 | 5991–6131 | 2.58–2.70 | 54.7–57.5 | 1638–1859 | 8.97–20.92 |
| 9 | 418.8–517.7 | 7.48–8.07 | 6131–6316 | 2.70–2.82 | 57.5–61.2 | 1859–2080 | 20.92–49.21 |
| 10 | | 8.07–10.17 | | | 61.2–66.2 | 2080–2301 | |

| Category\Factor | Slopa | Soilt | DNR (GI-9) | DNC (GI-9) | Popd (GI-10) | GRP (EI-9) | Livstd (QU-10) |
|---|---|---|---|---|---|---|---|
| 1 | Flat ground | Luvisols | 0–1.44 | 0–19.5 | 0.98–1.95 | 1.04–3.63 | 20.05–29.85 |
| 2 | North | Semi-luvisols | 1.44–2.35 | 19.5–31.9 | 1.95–3.79 | 3.63–6.22 | 29.85–52.72 |
| 3 | Northeast | Caliche soils | 2.35–3.79 | 31.9–39.8 | 3.79–7.28 | 6.22–8.81 | 52.72–55.99 |
| 4 | East | Arid soils | 3.79–6.11 | 39.8–44.8 | 7.28–13.89 | 8.81–11.39 | 55.99–69.06 |
| 5 | Southeast | Desert soils | 6.11–9.83 | 44.8–52.8 | 13.89–26.42 | 11.39–13.99 | 69.06–78.86 |
| 6 | South | Skeletol primitive soils | 9.83–15.79 | 52.8–65.2 | 26.42–50.17 | 13.99–16.57 | 78.86–111.53 |
| 7 | Southwest | Semi-hydromorphic soils | 15.79–25.35 | 65.2–84.6 | 50.17–95.20 | 16.57–19.16 | 111.53–157.27 |
| 8 | West | Hydromorphic soils | 25.35–40.68 | 84.6–115.1 | 95.20–180.56 | 19.16–21.75 | 157.27–186.67 |
| 9 | Northwest | Saline soils | 40.68–65.26 | 115.1–163.0 | 180.56–342.39 | 21.75–24.34 | 186.67–261.82 |
| 10 | | Anthrosols | | | 342.39–649.17 | | 261.82–853.17 |
| 11 | | Others | | | | | |

## 3. Results

### 3.1. Spatio-Temporal Variability of NDVI

The areas with mean growing season NDVI values of greater than 0.6 accounted for 5.34% of the entire area from 2000–2018 (Figure 7a), indicating the generally inferior nature of the vegetation cover in the Inner Mongolian grasslands. The spatial pattern of the multi-year mean NDVI during the growing season exhibited an increasing trend from south to north and from west to east (Figure 7a), which was highly consistent with the distribution pattern of water and heat resources. The northeastern part of the Inner Mongolian grasslands is located in the transitional zone between the Greater Khingan Range forest region and the Inner Mongolia temperate grasslands, and high NDVI values (>0.6) are concentrated in this area (Figure 7a). The topography of the central part of the region is dominated by high plains and low mountains, with good forage quality and NDVI values from 0.3–0.5. The western part of the region is subject to low rainfall and is home to xerophytic vegetation, which is mainly composed of xerophytic bunch grass mixed with semi-shrubs and allium plants. This area also has widely distributed low NDVI values (<0.2), indicating poor vegetation coverage (Figure 7a).

The growing season average NDVI of the entire study area ranged from 0.289 to 0.365 during the period of 2000 to 2018 (Supplementary Figure S2) and exhibited a significant increase at a rate of 0.003 a$^{-1}$ ($p < 0.05$). Figure 7b shows the NDVI changing trends at the pixel scale in the Inner Mongolian grasslands, which indicates that the NDVI mainly increased, with the areas of increase and the areas of decrease being 71.90 × 10$^4$ and 6.30 × 10$^4$ km$^2$, accounting for 91.94% and 8.06% of the entire region, respectively. The areas with significant increases in NDVI ($p < 0.05$) accounted for 45.63% of the entire region, and they were mainly distributed in the eastern part of the Mu Us sandy land (i.e., Erdos and Hohhot) and in the eastern areas of the Greater Khingan Range (i.e., Hinggan City, Tongliao City, and Chifeng City). Although the vegetation conditions improved in general, different degrees of degradation were also observed across the study area. The regions with

a decline in NDVI were mainly distributed in the central and western regions of the study area (Figure 7b), especially in four prefecture-level cities (i.e., Bayannur, Baotou, northern Ulanqab, and western Xilingol). The areas with significant decreases in NDVI were small, accounting for 0.44% of the study region, and they were relatively scattered.

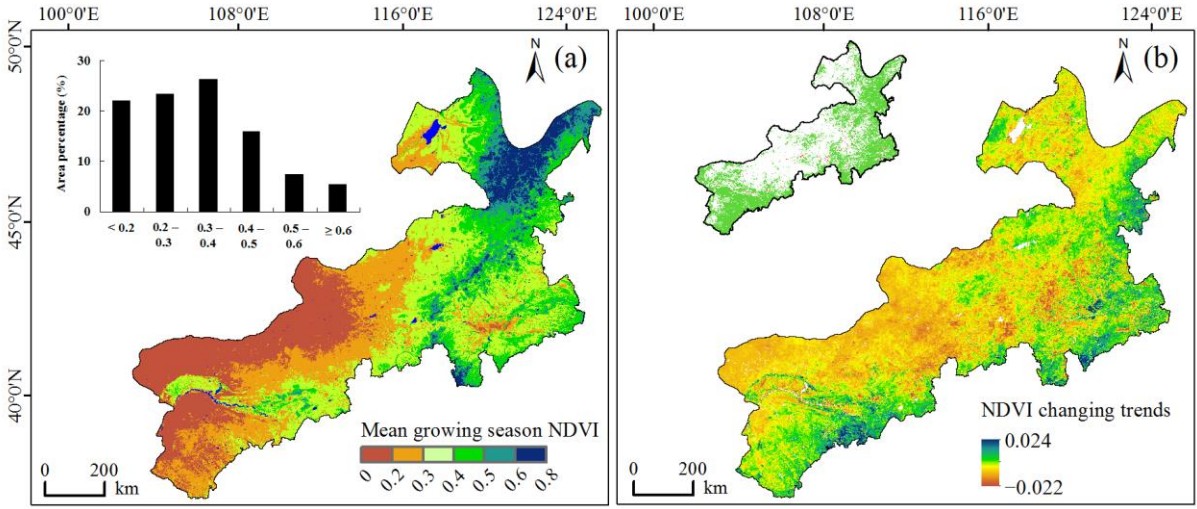

**Figure 7.** The (**a**) spatial distribution of the growing season average NDVI and (**b**) the slope of the change in the NDVI from 2000–2018. The inset graph in (**a**) is a statistical histogram; the inset map in (**b**) shows the significant decreases (red) and increases (green) in the NDVI at the 95% confidence level.

*3.2. Impacts of Natural and Human Factors on NDVI Changes*

3.2.1. Impacts of the 15 Factors on NDVI Changes

The *q*-statistics of all of the influencing factors passed the significance test ($p < 0.001$) (Supplementary Figure S3). The *q*-statistic values of the factors exhibited a marked difference that can be ranked as follows: Pre (0.217) > Livstd (0.182) > WS (0.173) > Popd (0.167) > GRP (0.126) > Alt (0.100) > Tem (0.096) > RH (0.088) > Soilt (0.067) > DNC (0.062) > SR (0.051) > LUCC (0.049) > DNR (0.035) > Slopd (0.015) > Slopa (0.001) (Supplementary Figure S3). These results indicate that the precipitation, which had the highest *q*-statistic value, predominantly explains the spatial heterogeneity of NDVI changes. The next most important factors were the livestock density, wind speed, and population density, with contributions of greater than 15%, while the impacts of the two topography factors (terrain slope and slope aspect) were relatively low, with *q*-statistic values of less than 0.02. Therefore, both the natural and human factors were identified as important factors influencing the vegetation NDVI changes in the Inner Mongolian grasslands.

3.2.2. Interactions between the 15 Factors

Two types of interaction relationships (i.e., nonlinear enhancement and bivariate enhancement) were identified among the 105 cases. For 55 cases, the *q*-statistics of the pairwise factor interactions were larger than the sum of the *q*-statistics of the two involved factors (Figure 8), which implies a nonlinear enhancement effect. The top five interactive *q*-statistics decreased in the following order: Pre∩Tem (0.393) > Tem∩Popd (0.336) > Pre∩Livstd (0.334) > Tem∩SR (0.332) > Tem∩RH (0.331). This indicates that the interactions between the climatic factors, population density, and livestock density had the greatest impacts on vegetation changes. These results show that the *q*-statistic of any pair of interacting factors was greater than the *q*-statistics of the single factors in the pair, implying that no factor influenced the vegetation changes in an independent manner but rather through interactions with the other factors.

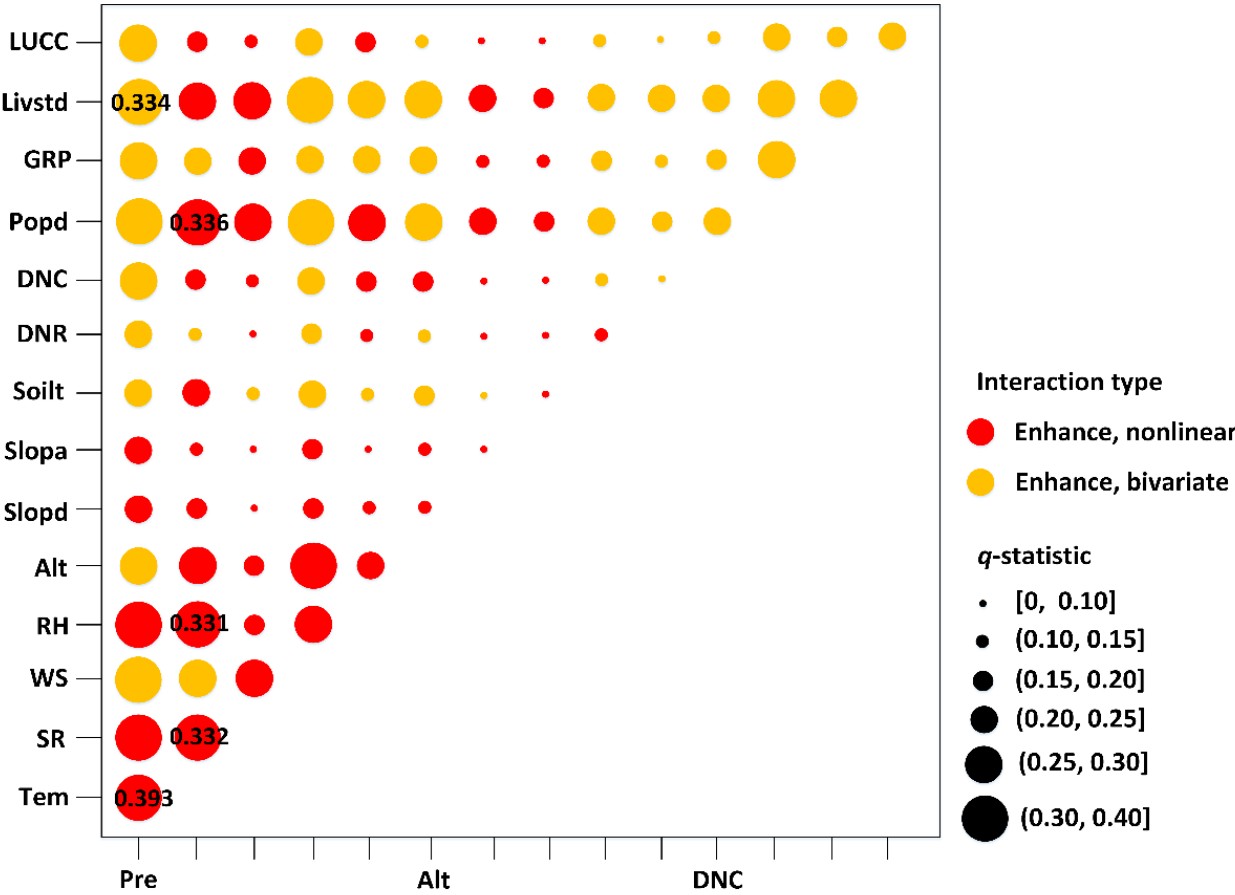

**Figure 8.** Influences of the interactions between two factors. Pre: precipitation; Tem: air temperature; SR: solar radiation; WS: wind speed; RH: relative air humidity; Alt: altitude; Slopd: slope; Slopa: slope aspect; Soilt: soil type; DNR: distance to the nearest road; DNC: distance to the nearest county centers; Popd: population density; GRP: Per capita gross regional product; Livstd: livestock density; and LUCC: land use/cover change type.

### 3.2.3. Effects of the Different Grades of the Factors

The rate at which the NDVI increased varied substantially with the different levels of the factors (Figure 9). Specifically, as the precipitation, population density, per capita gross regional product, and livestock density increased, the magnitude of the increase in the NDVI generally increased. As the wind speed increased, the rate of the increase in the NDVI generally decreased. The altitude, distance to the nearest road, and distance to the nearest county centers showed characteristics similar to those of the wind speed. As the relative air humidity increased, the rate of increase of the NDVI continued to increase and reached a maximum, and then it fluctuated slightly. The rate of increase of the NDVI fluctuated for different ranges or types of temperature, solar radiation, slope, aspect, and soil type (Figure 9). As shown in Table 3, most of the types of land use conversion led to an increase in the NDVI. The land use conversion from grasslands to croplands caused the largest increasing rate of NDVI. There were two types of land use conversions (from cropland to construction land and from unused land to construction land) that lead to a decrease in the NDVI.

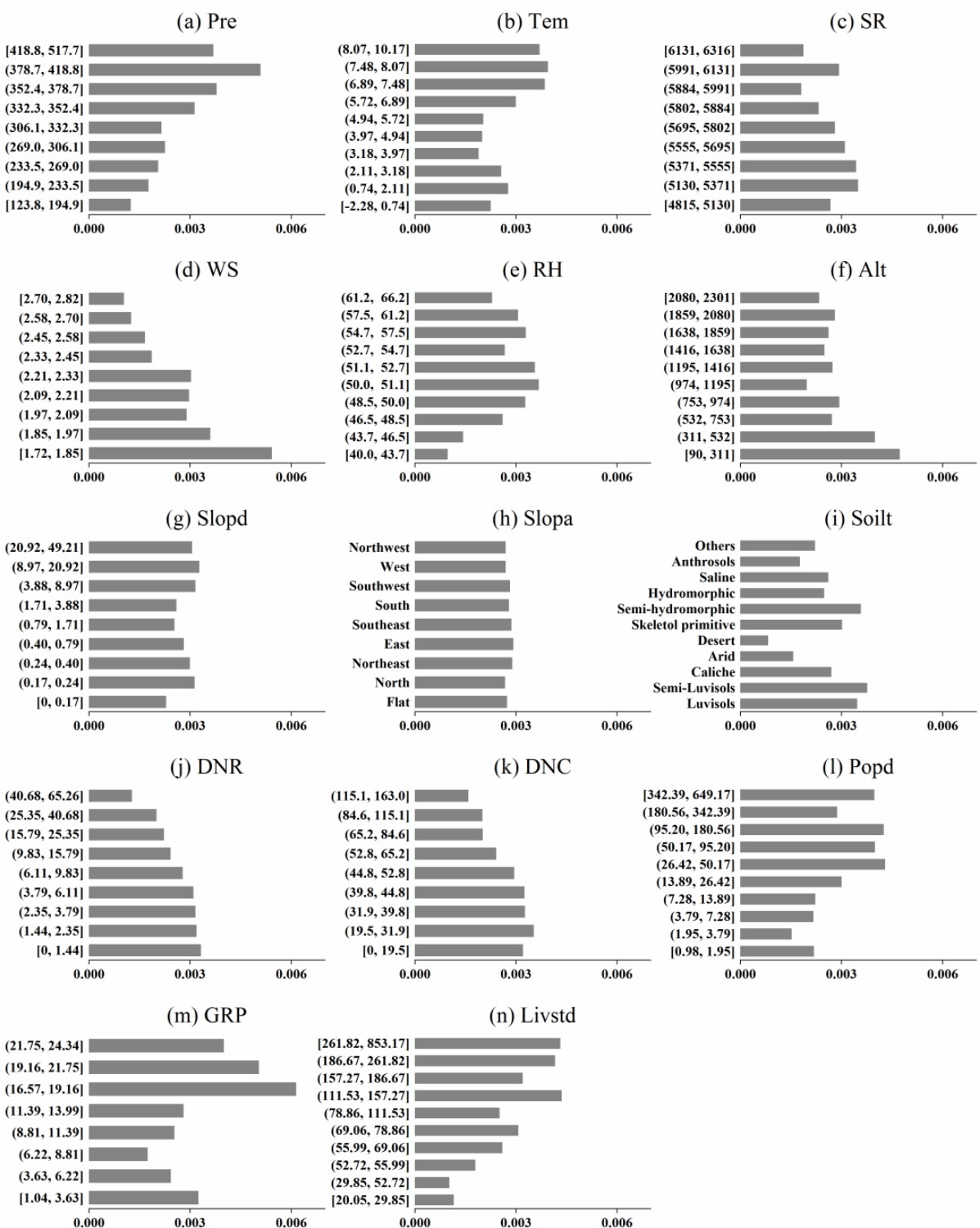

**Figure 9.** Influences of the factors' different grades on the magnitude of the increase in the NDVI. The units of Pre (precipitation), Tem (air temperature), SR (solar radiation), WS (wind speed), RH (relative air humidity), Alt (altitude), Slopd (slope), Slopa (slope aspect), DNR (distance to the nearest road), DNC (distance to the nearest county center), Popd (population density), GRP (per capita gross regional product), and Livstd (livestock density) are mm, °C, MJ·m$^{-2}$, m·s$^{-1}$, %, m, °, °, km, km, person·km$^{-2}$, 10,000 yuan person$^{-1}$, and sheep·km$^{-2}$, respectively. Soilt represents the soil type.

**Table 3.** Impacts of land use/cover change types on NDVI variations. The numbers in parentheses are the percentage of specific land use/cover change to the total area (%).

| 2000/ 2015 | Cropland | Forest | Grassland | Construction Land | Unused Land |
|---|---|---|---|---|---|
| Cropland | 0.0039 (13.649) | 0.0033 (0.080) | 0.0038 (0.229) | −0.0009 (0.106) | 0.0039 (0.020) |
| Forest | 0.0047 (0.036) | 0.0031 (9.461) | 0.0039 (0.059) | 0.0039 (0.013) | 0.0030 (0.014) |
| Grassland | 0.0057 (0.364) | 0.0037 (0.151) | 0.0025 (60.929) | 0.0010 (0.238) | 0.0026 (0.423) |
| Construction land | | | 0.0045 (0.006) | 0.0032 (1.381) | |
| Unused land | 0.0039 (0.048) | 0.0049 (0.038) | 0.0025 (0.474) | −0.0007 (0.047) | 0.0025 (12.227) |

## 4. Discussion

### 4.1. Applicability and Limitation of the GeoDetector Method

In relation to vegetation variations, numerous studies have explored the separation of natural and human factors. It should be noted that the commonly used methods (e.g., RESTREND, statistical correlation, or regression analysis) suffer from potential limitations. Specifically, RESTREND analysis cannot differentiate anthropogenic impacts from different aspects of human activities [14,18]. Statistical methods of evaluating the factors influencing vegetation changes mainly include correlation analysis [48], regression analysis [27], factor analysis [66], and geographically weighted regression [18]. However, these statistical methods involve assumptions regarding the data, fail to reveal the interactions between factors, or are hindered by the multicollinearities among the influencing factors [67]. The GeoDetector method was employed in this work, and it has three distinct advantages. (1) The GeoDetector method is not based on linear hypotheses, thus, it provides easier data preparation and wider applicability. (2) Working with both categorical and continuous variables, the GeoDetector method is not limited by data type. (3) The GeoDetector method can be used to determine how the interaction of explanatory variables affects the dependent variables without the restriction of multicollinearities [37,39,68,69]. Our study demonstrates that the GeoDetector method is an efficient technique for quantifying the impacts of driving factors and their interactions on vegetation changes.

Different grading standards (involving the discretization method and the stratification number) have certain impacts on the GeoDetector results [38,62,68]. However, the selection of the discretization methods and stratification numbers in prior model applications were subject to weaknesses such as randomness and subjectivity [39,59,64,70,71], which may introduce uncertainties and lead to misleading interpretations. In this study, an optimal discretization method was obtained based on the five types of unsupervised discretization methods (Figure 5). In addition, on the basis of a changing curve of the degree of influence (of influencing factors) with different numbers of stratifications, an optimal stratification number was also determined for each predictor variable (Figure 7). The optimization of factor-grading improves the accuracy and effectiveness of the modeling [38,63,72].

In this study, the socio-economic data are obtained from the statistical yearbooks at the county scale. It should be noted that the lack of spatial information of these socio-economic indicators forces them to be uniformly distributed within administrative divisions. This involves the spatial scale effect, which may have critical influences on the spatially stratified heterogeneity analysis. However, it has not been fully investigated and integrated in the GeoDetector method [38].

*4.2. Effects of Factors on Vegetation Changes*

4.2.1. Effects of the Main Natural Drivers

Our results indicate that precipitation was the dominant factor influencing the changes in the NDVI. This finding is consistent with the results of prior studies, which have indicated that vegetation growth in dryland ecosystems is very sensitive to precipitation changes [15,23,41,48]. As shown in Figure 9a, as the precipitation increased, the increase in the NDVI initially kept rising and reached a peak, and then it decelerated. A possible explanation for this is that the long-term cloud cover due to the excess precipitation may have resulted in reductions in temperature and solar radiation [48], which are not conducive to the improvement of the productivity of the grassland vegetation. In a similar vein, prior research reported that as precipitation increases, there is a threshold for the response of vegetation NDVI to precipitation, beyond which the magnitude of increase in NDVI driven by precipitation will decrease [73].

The rate of increase of the NDVI decreased substantially as the wind speed increased (Figure 9d) for two main reasons. First, the high wind speed increased evaporation and decreased the surface moisture, resulting in adverse effects on vegetation growth. Second, the Inner Mongolian grasslands are located in the sandstorm source region of northern China [46], with frequent strong winds in spring. In aeolian desertification areas, vegetation growth has been found to be constrained by burial and abrasion, the loss of surface soil resources, and the interruption of nutrient accumulation [74]. Zou and Zhai reported that vegetation coverage, as indicated by NDVI, was significantly negatively correlated with the occurrence frequency of spring dust storms in Inner Mongolia [75].

Overall, the magnitude of the increase in the NDVI values on the east-facing slopes was larger than that on the west-facing slopes (Figure 9h). One possible reason was that the study area is located in a marginal zone of the East Asian summer monsoon. Compared with the east-facing slopes, the west-facing slopes receive less precipitation and more solar radiation and thus are characterized by drier and hotter microclimates, which are harsher environments for vegetation growth.

The areas with semi-luvisols showed significant increases in the NDVI, while the desert soil had little effect on the increase in the NDVI (Figure 9i). Luvisol soils form under temperate forest and grassland vegetation. Due to the high accumulation of dead forest leaves and herbaceous debris and the cool climate, microbial decomposition is limited to a certain extent, leading to the formation of a humus layer with high fertility that can serve as important agricultural and forest soil resources. Desert soils, which develop under the temperate desert grassland vegetation in the northwestern marginal areas of the study region (Figure 4i), have little humus accumulation, a low organic matter content, and harsh environments (e.g., low precipitation, strong winds, and solar radiation), all of which are unfavorable for vegetation growth.

4.2.2. Effects of Human Activities

Livestock grazing is the main form of grassland utilization in Inner Mongolia [23,43]. Our results show that livestock density was the most influential anthropogenic factor affecting NDVI changes. Prior studies in arid Inner Mongolia were predicated on the belief that overgrazing substantially decreases vegetation cover and biomass production [76–78]. However, in this study, with an increase in livestock density, the magnitude of NDVI increase generally rose (Figure 9n), apparently suggesting, paradoxically, that grazing intensity, as indicated by livestock density, improved grassland vegetation. One possible reason for this is that the Inner Mongolia government has actively promoted the policy of herdsmen settlements and livestock pen-raising in recent decades [44]. According to the statistical yearbook of Inner Mongolia, in the past 19 years, the total area of livestock sheds has significantly increased at a rate of 9.31 million square meters per year ($p < 0.05$) (Figure 10), and the number of livestock in captivity has significantly increased at a rate of 5.37 million sheep units per year ($p < 0.05$) (Figure 10). During the same period, the total number of livestock in Inner Mongolia has increased at a rate of 2.07 million sheep

units per year, and the growth rate of livestock in captivity is much higher than that of the total amount of livestock. The intensive mode of livestock production accounts for the increasing proportion of animal husbandry production in pastoral areas. Due to the strong implementation of an ecological restoration policy and this intensive livestock production mode, forage sources are more dependent on external imports, and less damage is caused to the local vegetation. In addition, the livestock density is inherently high in areas with high vegetation coverage. Hence, with the increase in livestock density, grassland vegetation conditions still improved.

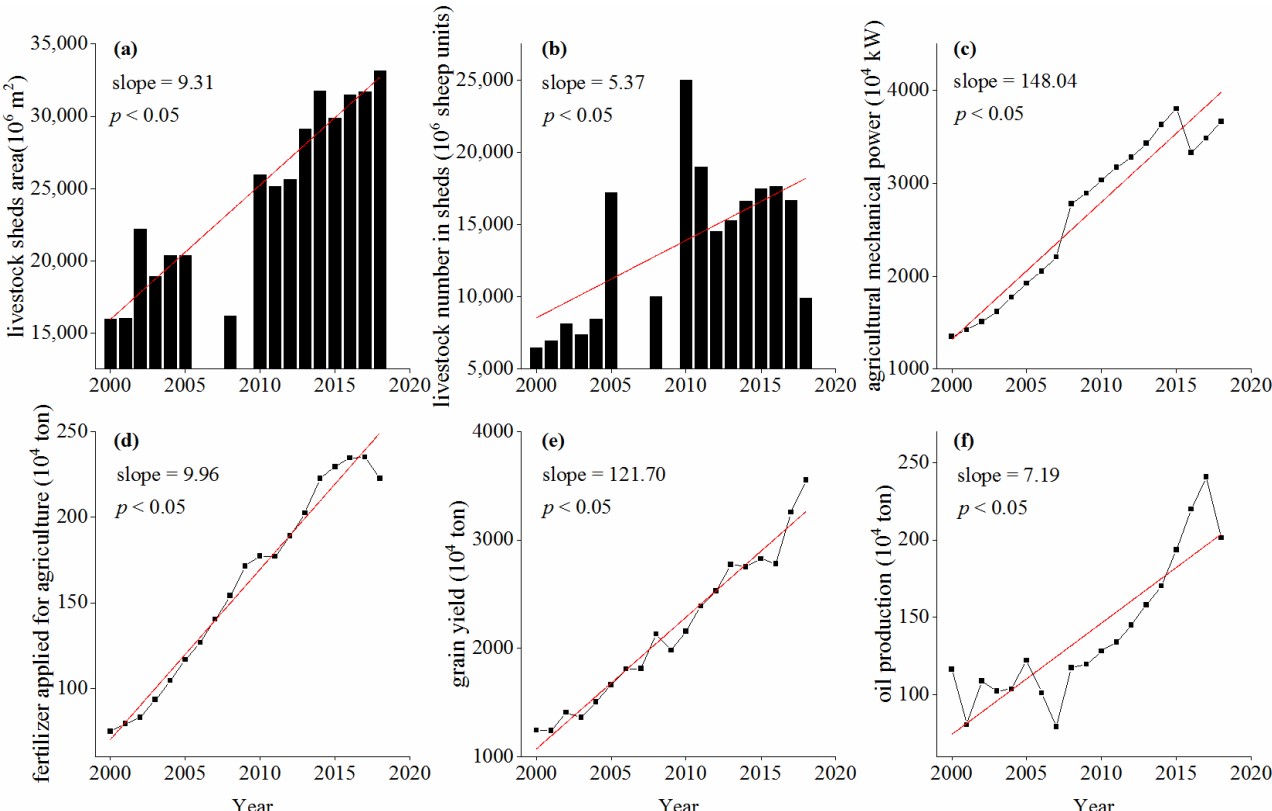

**Figure 10.** Inter-annual variations in (**a**) livestock shed area, (**b**) livestock numbers in sheds, (**c**) the total agricultural mechanical power, (**d**) the amount of fertilizer applied for agriculture, (**e**) the grain yield, and (**f**) the oil production in Inner Mongolia from 2000–2018. Note: The data were obtained from the Inner Mongolia Statistical Yearbook [79]. In (**a**,**b**), the data for livestock shed area and livestock numbers in sheds for 2006, 2007, and 2009 are not recorded in the Inner Mongolia Statistical Yearbook.

Although the GeoDetector method excludes the influence of multcollinearity among independent variables [37,39], the results show that the *q*-statistics and the ranking order of the livestock density, population density, and per capita gross regional product are very close (Supplementary Figure S3). This implied that the relationships between the human factors are closer than those between the natural factors. Actually, the interference of the anthropogenic factors in Inner Mongolia is not as complicated as those in developed areas. Grazing is the primary human activity affecting the ecological environment in the study area, and animal husbandry is the main source of income for local herders [41,44]. Where there are large numbers of livestock, there is often a high population density and social productivity (Figure 4j–n).

Land use/cover change is a manifestation of human activity [52]. Most of the types of land use conversion had positive influences on vegetation change in this study (Table 3). With the technological improvement and update of the industrial structures in agricultural

sectors, a 0.0039 a$^{-1}$ increase in the NDVI was observed in the unchanged cropland, which is corroborated by the fact that from 2000 to 2018, the agricultural mechanical power and fertilizer used for agriculture in Inner Mongolia increased by 2.71 and 2.98 times, respectively; the grain production increased from 12.42 × 10$^6$ t to 35.53 × 10$^6$ t, and the oil production increased from 1.16 × 10$^6$ t to 2.02 × 10$^6$ t [79] (Figure 10). The conversions of grassland and unused land into cropland increased the NDVI, indicating that reasonable reclamation has a positive effect on vegetation recovery. Through a series of ecological restoration measures, such as grazing prohibitions with grassland closures, restoration of cropland to grassland, and reforestation with hillside closures, the NDVI values of the unchanged grassland and forest land increased at rates of 0.0025 a$^{-1}$ and 0.0031 a$^{-1}$, respectively. For the unused land that was converted into grassland and forest, the NDVI values increased at rates of 0.0025 a$^{-1}$ and 0.0049 a$^{-1}$, respectively. For the cropland that was converted into grassland and forest, the NDVI values increased at rates of 0.0038 a$^{-1}$ and 0.0033 a$^{-1}$, respectively. Prior research also reported that the implementation of ecological restoration programs was beneficial to the improvement of vegetation coverage in Inner Mongolia [44,80]. However, urban expansion has caused decreases in the NDVI of 0.0009 a$^{-1}$ due to the conversion of cropland into construction land and 0.0007 a$^{-1}$ due to the conversion of unused land into construction land (Table 3). Therefore, more attention should be paid to ensuring the development of green infrastructures, such as parks and green spaces, during urban expansion.

The interactions among factors can greatly enhance the effect of a single factor (Figure 8). Although the distance to the nearest road, slope, and aspect did not contribute ideally to NDVI changes, their explanatory powers were enhanced when they interacted with other factors, especially precipitation and livestock density. Natural factors such as soil type ($q$(Soilt∩Livstd)) > $q$(Livstd)), slope ($q$(Slopd∩Livstd)) > $q$(Livstd)), and aspect ($q$(Slopa∩Livstd)) > $q$(Livstd)) tended to enhance the influence of human activities on vegetation changes.

*4.3. Limitations and Future Research Directions*

This study had certain limitations which can be improved in future research. First, the spatial differentiation of the relationships between NDVI variations and the driving factors was not taken into consideration. For example, most of the degraded vegetation from 2000 to 2018 was in the central and western regions of the Inner Mongolian grasslands (Figure 7b), and the interactions between the precipitation and temperature had the greatest impact on vegetation changes. Thus, with the high surface evaporation potential and low soil moisture due to a relatively small increase in precipitation and large increases in temperature in the central and western regions (Figure 11), the increase in the water stress level was probably the main reason for vegetation degradation. In further studies, the introduction of spatial statistical methods (e.g., a geographically weighted regression model), which can reflect the spatial nonstationarity of the parameters in different spaces, may improve our understanding of the spatial heterogeneity of the relationship between vegetation change and its driving forces [18,66]. Secondly, the drylands in northern China are regions with diverse land uses (mainly deserts and grasslands) and substantial seasonal climatic differences [81]. More evidence showed that soil moisture, which exhibits significant spatial and temporal variability [82], is crucial in regulating vegetation productivity. In future studies, a multiple time and spatial scale analysis can contribute to a better understanding of the drivers of vegetation growth change in order to develop suitable management schemes that are regionally and temporally specific. Thirdly, prior research observed the NDVI asymptotically saturating in high biomass regions [83]. Regarding this issue, the EVI (enhanced vegetation index) and SAVI (soil-adjusted vegetation index) were developed to make up for some of the shortcomings of the NDVI (e.g., atmospheric noise, soil background, saturation). Due to the limited spatial resolution of MODIS NDVI, it is difficult to meet the requirements of fine mapping. Combining the process with the Sentinel dataset or other vegetation indexes (e.g., EVI, SAVI) may help to obtain more

precise estimates of vegetation dynamics. Last but not least, if breakpoints, which indicate a shift in the mechanism of influence on the time series under certain circumstances, are neglected, the results of the trend analysis may lead to a misjudging of the factors that influence vegetation changes [40]. In future studies, the times at which breakpoints occurred should be first identified, noting points at which the time series was split into sub-series. Then, the trends and significance levels of the sub-series would be quantified separately to obtain more accurate conclusions regarding the driving forces of vegetation changes.

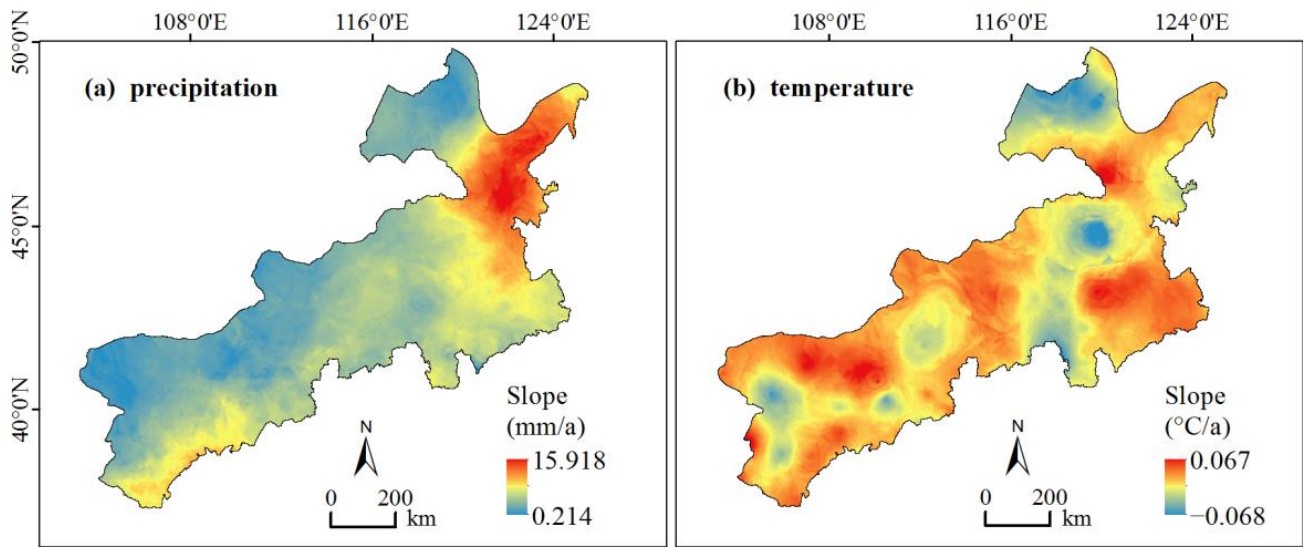

**Figure 11.** The slopes of changes in (**a**) precipitation and (**b**) temperature from 2000–2018.

## 5. Conclusions

In this research, we investigated the spatial and temporal variability in the mean growing season NDVI from 2000–2018 and quantified the individual and interactive influences of natural and human factors on NDVI change using the GeoDetector method in the Inner Mongolian grasslands. The results reveal that the NDVI increased at a rate of 0.003 $a^{-1}$. Both the natural and human factors had significant impacts on vegetation NDVI variations ($p < 0.05$), and the precipitation, livestock density, and wind speed had the greatest influences, while terrain slope and slope aspect had the lowest influences. The interactive impacts among factors often strengthened the impact of single factors.

Our study demonstrates that the GeoDetector method is an effective technique for disentangling the complicated driving factors of vegetation change. To effectively use the GeoDetector method, however, researchers need to carefully deal with the problem of spatial data discretization, which may introduce uncertainties and lead to misleading interpretations. The methodology used in this study can be applied to address the knowledge gap in the selection of the optimal discretization methods and the number of stratifications for further GeoDetector-based studies.

**Supplementary Materials:** The following supporting information can be downloaded at: https: //www.mdpi.com/article/10.3390/rs14143320/s1, Figure S1: Diagram illustrating the detection of interaction; Figure S2: Inter-annual variations in the mean growing season NDVI in the Inner Mongolian grassland from 2000–2018; Figure S3: The *q*-statistics of impact factors for NDVI changes.

**Author Contributions:** Conceptualization, S.L. and X.L. (Xiaobing Li); methodology, S.L., D.D. and X.L. (Xin Lyu); data curation, S.L., H.D. and X.L. (Xin Lyu); formal analysis, S.L., D.D. and H.D.; resources, X.L.; writing—original draft preparation, S.L., D.D. and X.L. (Xiaobing Li); writing—review and editing, S.L. and X.L. (Xiaobing Li); visualization, S.L. and H.D.; supervision, X.L. (Xiaobing Li) and J.G.; project administration, X.L. (Xiaobing Li) and J.G.; funding acquisition, X.L (Xiaobing Li) and J.G. All authors have read and agreed to the published version of the manuscript.

**Funding:** This research was funded by the Key Science & Technology Special Program of Inner Mongolia (No. 2021ZD0011 and 2021ZD0015) and the Project Supported by State Key Laboratory of Earth Surface Processes and Resource Ecology (No. 2022-ZD-02).

**Data Availability Statement:** All data that support the findings of the study are available from the corresponding author upon reasonable request.

**Acknowledgments:** We would like to offer our sincere thanks to those who participated in the data processing and manuscript revisions.

**Conflicts of Interest:** The authors declare no conflict of interest.

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
