# Peer review of "Quantitative Analysis of Natural and Anthropogenic Factors Influencing Vegetation NDVI Changes in Temperate Drylands from a Spatial Stratified Heterogeneity Perspective: A Case Study of Inner Mongolia Grasslands, China"

_remotesensing, doi:10.3390/rs14143320_

Round 1

Reviewer 1 Report

Your manuscript was well prepared and deals with a relevant subject. I present some suggestions below:

- If the study area mean temperature is -2 - 10 °C, there is probably no "period of high temperatures". Please correct that or cite the mean temperatures of the referred period.

- Please justify: "The 2000 and 2015 land cover type data (1000-m resolution) were retrieved..." - Why did you consider the period 2000 to 2018 if you have only maps up to 2015?

- Please clarify what is "artificial visual interpretation of Landsat thematic mapper images".

- What is "Geo-spatial data cloud"?

- How did you convert 26 categories into only 5 ( Cropland, Forest, Grassland, Construction land, and Unused land) classes?

- The section 3.2 is methodological, so it should come before the results section (Factor grading optimization in the GeoDetector method).

- Why did you chose 10 as the maximum number of stratification if few variables presented range of values for this category (table 2)?

- Did you notice "saturation" on the NDVI values? It can happen in situations when the vegetation is at its maximum vigour, such as some type of crops.

Reviewer 2 Report

Comments are attached below.

Reviewer 3 Report

see attached file

Round 2

Reviewer 2 Report

After going through the manuscript I feel that MS is significantly improved. I recommend acceptance.

Reviewer 3 Report

The revised version incorporates many changes and improvements.
The Geodetector method implementation is fully described.
Limitations are mentioned, as the need of using only discretized variables. Previous studies on application of Geo Detector method to landslides susceptibility mappingrevealed that the ranking orders of q statistics for
causative factors determined by Geo-detector under different discretization strategies are basically the same.
Results give a good illustration of spatial and temporal variability of grasslands in Inner Mongolia region.

This manuscript is a resubmission of an earlier submission. The following is a list of the peer review reports and author responses from that submission.